# Study on the Stability Mechanism of Peanut OBs Extracted with the Aqueous Enzymatic Method

**DOI:** 10.3390/foods12183446

**Published:** 2023-09-15

**Authors:** Chen Liu, Fusheng Chen

**Affiliations:** 1College of Biology and Food, Shangqiu Normal University, Shangqiu 476000, China; liuchen919258@163.com; 2College of Food Science and Engineering, Henan University of Technology, Zhengzhou 450001, China

**Keywords:** stability mechanism, peanut OBs, protein, phospholipid, interaction

## Abstract

In this study, the internal relationships among oil bodies (OBs), the protein–phospholipid interactions in aqueous phase, oil–water interface behavior, and the stability of reconstituted OBs were analyzed from the bulk phase, interface, and macro perspectives, and the stability mechanism of OBs was discussed. OB proteins and phospholipids were combined through hydrophobic and electrostatic interactions, resulting in the stretching of protein conformation. OB proteins and phospholipids act synergistically to increase interface pressure and the rate of increase in interface pressure with relatively stable elastic behavior, which is beneficial to the formation and stability of interfacial films. When OBs were reconstituted by an OB protein–phospholipid complex system, phospholipids bound to OB proteins through hydrophobic and electrostatic interactions. OB proteins and phospholipids uniformly covered the oil droplet surface of reconstituted OBs to form a stable interfacial film, which maintained the stability of OBs. The addition of phospholipids significantly reduced the particle size of OBs prepared by OB proteins in a dose-dependent manner, and particle size decreased with the increase in phospholipid content (*p* < 0.05). Phospholipids increased the net surface charge, enhanced electrostatic repulsion, and improved the physicochemical stability of reconstituted OBs. The stability mechanism elucidated in this study provides a theoretical basis for the demulsification of peanut OBs.

## 1. Introduction

Peanuts (*Arachis hypogaea*) are rich in protein and oil. Oil exists in peanuts in the form of oil bodies (OBs), which are composed of neutral lipids wrapped in phospholipids and proteins embedded in the phospholipid layer [1]. The proteins and phospholipids of OBs form a complex interfacial membrane between hydrophobic oil droplets and hydrophilic cytoplasm. Since OBs are generally exposed to various environmental stresses (e.g., high temperature, high pressure, and chemical and mechanical stresses), the strong OB membrane helps to maintain structural integrity under external influences [2,3]. Under neutral conditions, proteins and phospholipids constituting the membrane are negatively charged, conferring OBs with electrostatic repulsion and steric hindrance, thereby maintaining their stability and independence [4].

Aqueous enzyme extraction has emerged as a new technology for oil extraction with great application prospects owing to its numerous advantages, including product safety, low energy consumption, mild reaction conditions, the lack of the requirement of an organic solvent, and an environmentally friendly process [5]. However, OBs’ structure is stable, restricting the release of oil under the aqueous enzymatic method, representing a “bottleneck” in the industrial application of this method. Therefore, there is an urgent need for efficient new demulsification technology. Gaining a greater understanding of the detailed stability mechanism of OBs could provide a theoretical basis for their demulsification to establish an effective and economically efficient demulsification technology. To date, studies on the stability of OBs have mainly focused on the influence of OB composition, processing factors (e.g., temperature, pH, salt), and the microstructure of OBs. However, the influence of interactions between OB proteins and phospholipids and their interface behavior on the stability of OBs remains unclear.

In this study, we investigated the interaction between OB proteins and phospholipids in aqueous solution, along with their adsorption and rheological kinetics at the oil–water interface. OB proteins and phospholipids were used to reconstitute OBs, and the physicochemical parameters and microstructure of OBs were characterized. Internal relationships among the interactions between OB proteins and phospholipids in aqueous solution, the oil–water interface behavior, and the stability of reconstituted OBs were discussed, which can help to clarify the stability mechanism of OBs to provide a theoretical basis and guidance for developing an efficient demulsification method.

## 2. Materials and Methods

### 2.1. Materials

Peanut samples (Yuhua 23) were purchased from the Henan Academy of Agricultural Sciences (Zhengzhou, China). Viscozyme^®^ L was purchased from Novozymes (Novo, Copenhagen, Denmark). Peanut oil was purchased from Luhua Group Co. Ltd. (Laiyang, China). Florisil molecular sieve adsorbent (60–80 mesh) was purchased from Shanghai Macklin Biochemical Co. Ltd. (Shanghai, China). Nile Red and fluorescein isothiocyanate (FITC) were purchased from Sigma-Aldrich Co. (Shanghai, China). Rd-DHPE was purchased from Aladdin Biotechnology Co. Ltd. (Shanghai, China).

### 2.2. Samples Preparation

#### 2.2.1. Extraction of Peanut OBs

The aqueous enzymatic method reported by Zhou et al. [6] was used to extract peanut OBs, with some modifications. Twenty grams of skinless peanut seeds was added to deionized water, and then stored at 4 °C for 18 h. Peanuts were ground for 2 min using a C022E food processor (Joyoung Co., Ltd., Jinan, China). Viscozyme^®^ L (1.25%) was used to conduct enzymolysis at 50 °C for 2 h. The suspension was centrifuged for 20 min at 5000× *g* and the upper layer was considered the OB-a sample. The remaining water phase and residue were stirred well, and then incubated at 50 °C for 30 min. The upper layer was considered the OB-b sample. OB-a and OB-b are referred to as peanut OBs. 

#### 2.2.2. Extraction and Characterization of OB Proteins

Three volumes of n-hexane were added to peanut OBs and stirred for 4 h. Neutral fat was removed by centrifugation at 5000× *g* for 20 min. These steps were repeated three more times to ensure complete precipitation and fat removal. New chloroform–methanol solution (three volumes, 2:1 *v*/*v*) was added to the precipitate, the solution was stirred for 4 h, and phospholipids were removed by centrifugation at 5000× *g* for 20 min. Nitrogen blowing was used to remove residual organic solvent from the precipitate to obtain OB protein samples. Protein composition was analyzed according to a previously described method [7]. In brief, the protein powder was dissolved in reducing sodium dodecyl sulfate–polyacrylamide gel electrophoresis (SDS-PAGE) sample buffer to prepare a protein solution (2 mg/mL), which was placed in a boiling water bath for 5 min. Stacking gel (5%) and resolving gel (12%) were used for SDS-PAGE, and the initial current was 20 mA, which was increased to 40 mA when the protein entered the resolving gel. The gel was fixed with stationary liquid for 30 min after electrophoresis, and then stained for 1 h. Finally, the gel was decolorized until it was clear.

#### 2.2.3. Extraction and Characterization of Phospholipids

Phospholipids were extracted using the hydration degumming method, as reported previously [8]. One hundred grams of crude oil was placed in a beaker in a water bath, and then citric acid (45 g/100 mL, 222 μL) was added once the crude oil temperature reached 80 °C. The mixture was homogenized at 23,000× *g* for 30 s and magnetized at 80 °C for 30 min at 200 rpm. Deionized water of 85 °C (2 mL/100 g) was added, and the water bath temperature was increased to 85 °C. When the temperature of the mixture reached 85 °C, stirring was continued for 20 min, followed by slow stirring for a further 30 min. Finally, the solution was centrifugated at 5000× *g* for 10 min, and the lower precipitate was taken as a phospholipid sample. Phospholipids were purified by cold acetone and centrifuged at 4 °C and 5000× *g* for 20 min, and these steps were repeated three times. Phospholipids were determined by ^31^P nuclear magnetic resonance (NMR) spectroscopy (HD500 MHZ, Bruker, Bremen, Germany). In brief, 200 mg of the phospholipid was mixed with 0.5 mL methanol, 0.5 mL ethylenediaminetetraacetic acid (0.2 mol/L, pH 7.0), and 0.5 mL deuterated chloroform (CDCl_3_) containing 5 mg triphenyl phosphate (dissolved in 50 mL CDCl_3_) successively, and then centrifuged at 4000× *g* for 5 min. The lower fluid (0.5 mL) was carefully drawn and ^31^P NMR was run at a probe temperature of 29 °C, pulse width of 22 μs, sweep width of 9718 Hz, acquisition time of 1–2 s, and interval of 10 s, with 256 scans [9].

#### 2.2.4. Purification of Peanut Oil

Commercial peanut oil contains certain surfactants and other impurities, which need to be purified before use. Florisil molecular sieve adsorbent (60–100 mesh, 2%) was added to peanut oil and stirred for 1 h, followed by centrifugation at 5000× *g* for 10 min to remove the absorbent. This step was repeated three times, and the purified peanut oil was stored at 4 °C in the dark until use [10].

### 2.3. Interaction Analysis between OB Proteins and Phospholipids in the Aqueous Phase

#### 2.3.1. Preparation of OB Proteins, Phospholipids, and OB Protein–Phospholipid Complex Solutions

A certain mass of OB proteins and phospholipids were dissolved in phosphate buffer (10 mM/L, pH 7.0) to prepare the OB protein solution (0.5%), phospholipid solution (0.05%, 0.1%, 0.2%), and OB protein–phospholipid complex solution at each phospholipid concentration. The solution was stirred at room temperature for 3 h, and then stored overnight at 4 °C.

#### 2.3.2. Endogenous Fluorescence Spectrum Analysis

The effects of phospholipids on the OB protein molecules were evaluated using endogenous fluorescence spectroscopy to reflect their interaction. Endogenous fluorescence spectra of the protein and protein–phospholipid complex solutions were detected at 25 °C. The spectrum in the range of 300–400 nm was scanned at an excitation wavelength of 290 nm and an excitation and emission slit width of 5 nm.

#### 2.3.3. Isothermal Titration Calorimetry (ITC)

ITC is a powerful tool to characterize molecular binding thermodynamics. By detecting the thermal effect of the interaction between two solutions, the thermodynamic parameters of the interaction can be obtained to characterize the interaction between two molecules [11,12]. OB protein solution (2 mg/mL) and phospholipid solution (20 mg/mL) were prepared with phosphate-buffered solution (10 mM/L, pH 7.0) and stirred overnight. Phospholipid solution (50 μL) was injected with OB protein solution (250 μL) into the sample pool. The syringe was automatically titrated 25 times consecutively, each time titrating 2 μ with a titration interval of 180 s, temperature of 25 °C, and stirring speed of 300 rpm during the titration process. The thermodynamic parameters related to the interaction between OB proteins and phospholipids were determined with a one-site independent binding model using TA Nano Analyzer software, including the binding constant (Kd), reaction enthalpy change (ΔH), reaction entropy change (ΔS), and reaction binding ratio (n).

#### 2.3.4. Determination of Zeta Potential

The zeta potential of OB proteins, phospholipids, and OB protein–phospholipid complex solutions was measured using a zeta potential analyzer (Zetasizer Nano ZSP, Marvin Instrument Co., Ltd., Marvin, UK).

### 2.4. Interface Characteristic Analysis

#### 2.4.1. Dynamic Interfacial Tension Measurement

Dynamic drop shape analysis was used to detect the change in interface pressure at the oil–water interface with adsorption time, according to the method of Wang et al. [13] with appropriate modifications. The needle tip of the syringe loaded with sample to be tested was immersed into a transparent glass tank containing purified peanut oil, so that 10 μL droplets with a complete shape formed on the tip. Shape images of droplets were periodically collected by a high-speed video camera system, and were used to calculate the interfacial tension (γ). The determination of each sample was carried out for 180 min at 25 °C. Interface pressure (*π*, mM/m) was calculated according to Equation (1):(1)$$\pi =\gamma_{0}-\gamma$$
where γ_0_ is the surface tension of phosphate buffer (10 mM/L, pH 7.0) and γ is the surface tension of the sample to be tested.

When the protein concentration is low, diffusion is mainly driven by the concentration gradient, and adsorption kinetics are controlled by the diffusion process. The relation of interface pressure and adsorption time can be converted into Equation (2) using the Ward and Tordai equations [14,15].
(2)$$\pi =2C_{0}KT{\left(Dt/3.14\right)}^{1/2}$$
where C_0_ is the concentration of the continuous phase, K is the Boltzmann constant, T is the absolute temperature, and D is the diffusion coefficient. 

When diffusion controls the adsorption kinetics, the *π*~t^1/2^ curve is linear, and the slope of the linear part can be used to characterize the diffusion rate (K_diff_). Interface pressure was fitted using the first-order Equation (3), which can be used to calculate the penetration rate (Kp)and rearrangement rate (Kr) of the surfactant at the oil–water interface [16].
(3)$$ln\left[\left(\pi_{f}-\pi_{t}\right)/\left(\pi_{f}-\pi_{0}\right)\right]=-k_{i}t$$
where *π*_f_, *π*_0_, and *π*_t_ are the interface pressure at 10,800 s, 0 s, and any time, respectively; k is the first-order rate constant.

#### 2.4.2. Interface Rheological Properties Analysis

Changes in viscoelastic modulus (*E*), elastic modulus (*E_d_*), viscous modulus (*E_v_*), and phase angle (θ) with the adsorption time of samples at the oil–water interface were detected by dynamic drop shape analysis and the oscillating drop technique according to the methods of Wang et al. [13] and Zhu et al. [17], respectively, with appropriate modifications. The sample solution and purified peanut oil were placed in a syringe and transparent glass tank, respectively. A stainless steel needle was inserted into the peanut oil, formed a 10 uL droplet, and was then balanced for 60 s. Droplets were generated with periodic sinusoidal oscillation and images of the droplet shape were collected periodically by a video camera system for analysis. The oscillation frequency (*f*) was held constant at 0.1 Hz, the amplitude (ΔA/A0) was set to 10%, and the test temperature was 25 °C.

### 2.5. Preparation of Reconstituted OBs

Peanut OBs were reconstituted with different additive amounts of OB proteins and phospholipids (see Section 2.3.1) according to the method of Cai Y et al. [18] with appropriate modifications. Purified peanut oil (10%) was added into the OB protein solution, phospholipid solution, or OB protein–phospholipid complex solution. Reconstituted OBs were obtained by high-speed shearing for 1 min at 10,000 r/min, followed by high-pressure homogenizing at 50 MPa for 1 min.

### 2.6. Determination of Particle Size and Zeta Potential of Reconstituted OBs

Reconstituted OBs were diluted to 1% using phosphate-buffered solution and left to stand for 24 h. Subsequently, particle sizes of evenly dispersed mixtures were measured by a laser particle size analyzer (BT-9300H, Dandong Baxter Instrument Co., Ltd., Dandong, China). The zeta potential of reconstituted OBs diluted to 0.1% using phosphate buffer was measured using the Zetasizer Nano ZSP system as described above.

### 2.7. Microstructure of Reconstituted OBs 

The microstructure of the reconstituted OBs was evaluated by confocal laser scanning microscopy (CLSM), referring to Sui et al. [19] and Anant et al. [20], with some modifications. Reconstituted OBs were diluted with deionized water to prepare 10% OB emulsion. OB emulsions (2 mL) were mixed with 10 μL Nile Red and FITC, and then dropped onto a fluted slide and covered with a cover glass. In addition, OB emulsions (2 mL) were mixed with 10 μL Rd-DHPE alone and then dropped onto a fluted slide and covered with a cover glass. The microstructure of OBs was observed by CLSM (FV 3000, Olympus Corporation, Tokyo, Japan).

### 2.8. Statistical Analysis

All measurements were repeated at least three times. Representative SDS-PAGE and CLSM images are provided. Data were compared using one-way analysis of variance and Tukey’s post hoc test assessed at a significance level of *p* < 0.05 using SPSS. Images were processed in Origin software.

## 3. Results and Discussion

### 3.1. Characterization of OB Proteins and Phospholipids

The protein compositions of peanut OBs and extracted OB protein powder are shown in Figure 1A. A previous study [21] showed that the inherent proteins of OBs are composed of oleosin, caleosin, and steroleosin. However, protein bodies are destroyed in the grinding process, resulting in the release of a large number of proteins. Some proteins adsorbed to the OB surface during the aqueous enzymatic extraction process were identified, including lipoxygenase, arachin, conarachin, allergen, and ferritin. The subunit compositions of the extracted OB protein powder (lane 2) were consistent with those of OB proteins (lane 1), and the intensity of bands with the same molecular weight was similar for the two samples, indicating that the extraction process of OB protein powder did not affect the overall protein composition. Phospholipid molecules have both polar regions and nonpolar lipid tails, the former being oriented to the aqueous phase, whereas the latter forms a hydrophobic phase [22]. Phospholipids are an important factor contributing to the stability of OBs. The ^31^P NMR spectra showed that phospholipids of OBs were mainly composed of phosphatidyl choline (PC), phosphatidyl inositol (PI), phosphatidyl ethanolamine (PE), and phosphatidic acid (PA), with a relative content of 54.87%, 23.04%, 16.76%, and 5.33%, respectively (Figure 1B). These results are similar to those reported by Zhao et al. [23] and Schiller et al. [24].

### 3.2. Effect of Phospholipids on the Endogenous Fluorescence Spectrum of OB Proteins

Fluorescence spectroscopy is a sensitive technique that is used to reflect changes in a protein’s tertiary structure. Using a 290 nm excitation wavelength, fluorescence spectra focus on tryptophan as the main emitter group, reflecting changes in the polarity of the tryptophan microenvironment. The redshift of the maximum fluorescence peak in the endogenous fluorescence spectrum indicates that hydrophobic groups buried inside a protein molecule are more exposed to the solvent, suggesting that tryptophan residues in the protein microenvironment are more polar, while the blueshift of the maximum fluorescence peak indicates that emitter groups are located in a more hydrophobic microenvironment [25,26]. Accordingly, the change in the fluorescence spectrum can reflect the influence of phospholipids on the microenvironment of fluorescently labeled groups inside protein molecules, thereby reflecting changes in the protein structure. As shown in Figure 2, the addition of phospholipids significantly reduced the maximum fluorescence intensity of the OB proteins, which gradually decreased with an increase in the phospholipid concentration. Such fluorescence quenching indicated an interaction between OB proteins and phospholipids. The maximum fluorescence peaks of all samples ranged from 348 to 350 nm, which was outside the 290 nm range, indicating a blueshift and that the tryptophan residues in all samples were in a hydrophobic environment [27]. With an increase in the added phospholipid content from 0% to 0.2%, the maximum fluorescence peak showed a slight blueshift from 350 nm to 348 nm, indicating that the hydrophobicity of the microenvironment around tryptophan residues on the protein peptide chain increased, possibly due to an interaction between tryptophan and the hydrophobic portion of the phospholipids.

### 3.3. Interaction between OB Proteins and Phospholipids

ITC is widely used to monitor the kinetic process of interactions between molecules, and can further help to identify the reaction type and strength of interaction according to thermodynamic parameters (ΔH, ΔS, Kd and n) [28]. The thermal change between OB proteins and phospholipids based on ITC analysis is shown in Figure 3A. When phospholipid solution was dropped into OB protein solution, an obvious heat flow peak appeared, indicating phospholipids bound to OB proteins. Peak height decreased with the increase in phospholipids, possibly due to the gradual saturation of the number of binding sites on OB proteins. The titration curve was fitted through a one-site independent combination model, and the corresponding thermodynamic parameters are shown in Figure 3B. The binding ratio n of the reaction was 10, indicating that protein molecules could bind 10 phospholipid molecules. Based on thermodynamic laws, the interaction between macromolecules and small molecules can be interpreted as follows [29,30]: when ΔH < 0, interactions are mainly electrostatic interactions, van der Waals forces, or hydrogen bonding; when ΔS > 0, only electrostatic interactions are present; when ΔS < 0, van der Waals forces or hydrogen bonding are the main interactions; and when ΔH > 0, ΔS > 0, hydrophobic interactions dominate. During the reaction between OB proteins and phospholipids, ΔH > 0 and ΔS > 0, indicating that interaction was driven by entropy change, and was mainly characterized by hydrophobic interactions, which was consistent with the results in the endogenous fluorescence spectrum.

### 3.4. Effect of Phospholipids on the Zeta Potential of OB Proteins

Zeta potential reflects the charged properties and strength of the electric potential between droplets. The higher the absolute value of zeta potential, the greater the electrostatic repulsion between droplets [8]. The effect of phospholipids on the zeta potential of OB protein solution is shown in Figure 4. Under neutral conditions, both the OB protein and phospholipid solutions were negatively charged. The zeta potential of the OB protein solution was −26.17 mV, and the absolute value of the zeta potential of phospholipid solution increased significantly with the increase in phospholipid concentration. The absolute value of zeta potential was significantly higher than that of the OB protein solution (*p* < 0.05). The addition of phospholipids significantly increased the absolute value of the zeta potential of OB protein solution (*p* < 0.05), and the electronegativity of the OB protein–phospholipid complex solution increased with the increase in phospholipid concentration (*p* < 0.05). OB proteins are negatively charged under neutral conditions; however, amphiphilic fragments of oleosin [31] and caleosin [32] contain relatively rich basic amino acids, which could interact with negatively charged groups of phospholipids through electrostatic interaction [33]. Therefore, the zeta potential absolute value of the phospholipid solution was significantly decreased due to negatively charged OB proteins (*p* < 0.05). Considering the results of the endogenous fluorescence spectrum and ITC, in addition to zeta potential analyses, phospholipids were bound to OB proteins through hydrophobic interactions, which induced the unfolding of protein structure, thereby exposing more basic amino acids to promote electrostatic interactions between OB proteins and phospholipids, leading to the increase in zeta potential absolute value of OB protein solution (*p* < 0.05). Therefore, OB proteins and phospholipids bind through both hydrophobic and electrostatic interactions in aqueous solution.

### 3.5. Adsorption Behavior of OB Proteins and Phospholipids at Oil–Water Interface

Changes in interfacial pressure when protein is adsorbed onto the oil–water interface can be used to characterize the adsorption process [34]. The dynamic effects of phospholipids added at different concentrations on the interfacial pressure of OB proteins at the oil–water interface with adsorption time is shown in Figure 5. The interfacial pressure of all samples increased gradually with the increase in adsorption time, indicating that OB proteins and phospholipids were gradually adsorbed onto the oil–water interface. After adsorption for 10,800 s, interfacial pressure continued to increase but more slowly, indicating that the interface had not yet reached the adsorption equilibrium state. Proteins and other macromolecular surfactants have a weak ability to reduce interfacial tension and thereby adsorb slowly at the interface; therefore, even when these molecules are adsorbed over a longer time (2–3 days) at the oil–water interface, it is difficult to achieve equilibrium [35]. Compared with those of the OB protein system alone, the interfacial pressure and its rate of increase were significantly higher for the OB protein–phospholipid complex system, which indicated that OB proteins and phospholipids have a synergistic effect in increasing the interfacial pressure. Therefore, the hydrophobic and electrostatic interactions of phospholipids with OB proteins change the structure of OB proteins, conferring a beneficial effect on the adsorption of the proteins at the interface.

The adsorption of proteins from the bulk phase to oil–water interface involves three main steps: diffusion, permeation, and rearrangement [34]. Adsorption kinetics and the corresponding parameters of the OB protein solution and OB protein–phospholipid complex solution at the oil–water interface are shown in Figure 6 and Table 1, respectively. At the initial stage of the increase in interfacial pressure, the adsorption of proteins on the oil–water interface is dominated by diffusion. There is a linear relationship between *π* and t^1/2^, and the slope of the curve represents the diffusion rate (K_diff_) [36]. The curves in Figure 6B all include two linear regions, in which the first slopes of the curves are the first-order permeation constant, namely, the permeate rate (Kp), and the second slopes of the curves are the first-order rearrangement constant, namely, the rearrangement rate (Kr) [37]. As shown in Table 1, the K_diff_ of OB proteins at the oil–water interface showed a trend of first increasing and then decreasing with the increase in phospholipid additive content. Compared to that of OB protein solution, K_diff_ increased significantly when phospholipid additive content was 0.05% and 0.1% (*p* < 0.05), indicating that interaction between phospholipids and OB proteins promoted the expansion and diffusion of OB proteins at the oil–water interface to a certain extent. However, the K_diff_ of OB proteins decreased with the addition of 0.2% phospholipids, which might have been due to an increase in the viscosity of system, resulting in a decreased migration rate of OB proteins. The Kp and Kr values of OB proteins decreased with the increase in phospholipid content, and it was also possible that the phospholipids increased the viscosity of the system to restrict the movement of OB proteins at the oil–water interface. In addition, the density of OB proteins per unit area on the interface increased with the quantity of proteins absorbed on the interface, consequently limiting the expansion and rearrangement of OB proteins at the oil–water interface [38].

### 3.6. Rheological Properties of OB Proteins and Phospholipids at Oil–Water Interface

Figure 7 shows the effects of phospholipids on the interface expansion characteristic parameters (*E*, *E_d_*, tan θ) of OB proteins at the oil–water interface. The *E* of all samples increased with the increase in adsorption time, indicating that the formed adsorption layer was viscoelastic, which was attributed to the adsorption and intermolecular interaction between OB proteins and phospholipids at the oil–water interface. In addition, the *E_d_* and *E* values of all samples were similar in magnitude and variation trend, further indicating that the interface film was mainly elastic. Compared with OB proteins alone, the addition of phospholipids significantly reduced *E_d_*, because interactions between phospholipids and OB proteins could destroy interactions between protein molecules, weaken the interface membrane structure, and reduce its viscoelasticity [39,40]. The *E-π* curve can reflect the amount of protein adsorption at the interface and the degree of interaction between protein molecules, and its slope reflects the equilibrium state of adsorbed substances at the oil–water interface [41]. As shown in Figure 7B, *E* increased with the increase in *π* for individual OB protein samples, indicating that the adsorption capacity of OB proteins on the oil–water interface gradually increased, and the interaction between molecular residues of adsorbed protein existed and were enhanced [42]. Similarly, the *E* of the OB protein and phospholipid complex system also increased with the increase in *π*, indicating the absorption of OB proteins and phospholipids on the oil–water interface, and the interactions between OB proteins and other OB proteins and OB proteins and phospholipids adsorbed on the interface gradually strengthened. The *E-π* curve slope of the OB protein sample was 2.21, which was not an ideal adsorption state (the slope of the *E-*π curve was 1 in an ideal state), indicating strong intermolecular interactions between OB proteins adsorbed on the oil–water interface. For all OB protein–phospholipid complex systems, the slope of the *E-π* curve was greater than 1, which was also not an ideal adsorption, and further suggesting strong interaction between components in the adsorption layer. When the additive content of phospholipids was 0.05%, the slope of the *E-π* curve was close to that of the OB protein sample alone, indicating that interactions between OB protein molecules dominated the interface. The slope of the *E-π* curve decreased with the increase in phospholipid content, because interactions between OB proteins and phospholipids were enhanced and interactions between protein molecules at the interface were weakened. As shown in Figure 7D, the tangent value of the phase angle (tan θ) of OB proteins decreased with the increase in adsorption time and was always less than 1, indicating that the elastic behavior of interface was gradually enhanced, and the degree of interfacial protein crosslinking caused by the molecular rearrangement of OB proteins at the interface was increased [43,44]. With the increase in phospholipid content from 0.05% to 0.2%, tan θ decreased with the increase in adsorption time, which was also less than 1, and the interface showed steady elastic behavior.

### 3.7. Stability and Microstructure of Reconstituted OBs

The effects of OB proteins and phospholipids on the stability of reconstituted OBs were evaluated by particle size and zeta potential measurements. A larger absolute value of zeta potential indicates a higher charge on the surface of oil droplets, greater electrostatic repulsion between oil droplets, and less aggregation, while a smaller particle size indicates a more stable system [45,46]. As shown in Figure 8, the particle size of reconstituted OBs stabilized by OB proteins was larger, and its d_43_ value was 6.73 µm. As a small-molecule surfactant, phospholipids are more surface active than OB proteins and can reduce interfacial tension more effectively [47]. The particle size of reconstituted OBs stabilized by phospholipids alone was significantly lower than that of OBs reconstituted with OB proteins (*p* < 0.05). For reconstituted OBs prepared by OB proteins and phospholipids, the addition of phospholipids decreased the particle size of reconstituted OBs stabilized by OB proteins, and the d_43_ value decreased significantly with the increase in phospholipid content (0.05%, 0.1%, and 0.2%), corresponding to d_43_ values of 4.78, 3.64, and 2.93 µm, respectively. 

As shown in Figure 8B, reconstituted OBs prepared by OB proteins or phospholipids alone were negatively charged, with the latter OBs showing a significantly higher absolute value of the charge than the former group (*p* < 0.05). For reconstituted OBs prepared with OB proteins and phospholipids, OB proteins were bound to phospholipids through electrostatic and hydrophobic interactions, phospholipids increased the negative charge of reconstituted OBs prepared by OB proteins, the absolute value of zeta potential increased significantly with the increase in phospholipid content (*p* < 0.05), and the stability of reconstituted OBs increased. According to the analysis of particle size and zeta potential, the addition of phospholipids increased electrostatic repulsion, decreased particle size, and increased the stability of reconstituted OBs, and the stability increased with the increase in phospholipid content. Therefore, phospholipids play an important role in the stability of reconstituted OB emulsions. Consistently, Deleu et al. [48] found that the net charge density on the surface of reconstituted OBs increased with the increase in phospholipid content, and phospholipids had a significant influence on the stability of reconstituted OBs.

Finally, the microstructure of reconstituted OBs was observed by CLSM, and the results are shown in Figure 9. As shown in Figure 9A,D,F,H, oil droplets in peanut OBs and reconstituted OBs all emitted red fluorescence, and were spherical and evenly dispersed in the water phase. The surface of oil droplets of reconstituted OBs prepared by phospholipids showed uniform blue fluorescence (Figure 9E), indicating that the phospholipids were uniformly distributed on the surface of the oil droplet. Similarly, the proteins were uniformly distributed on the surface of reconstituted OBs prepared with OB proteins (Figure 9G). During the preparation process of reconstituted OBs, OB proteins and phospholipids quickly adsorbed to the oil–water interface. In addition, OB proteins interacted with phospholipids by hydrophobic and electrostatic interaction, which promoted the adsorption of OB proteins and phospholipids at the oil–water interface, and was beneficial to the formation and stability of the oil–water interface. As shown in Figure 9I,J, OB proteins and phospholipids were both distributed on the oil droplet surface uniformly. The particle size of reconstituted OBs prepared by OB proteins was clearly larger than that of reconstituted OBs prepared by phospholipids, which was consistent with the particle size analysis. The content of OB protein and phospholipid in peanut and reconstituted OBs was low, but proteins and phospholipids uniformly covered the surface of oil droplets and interacted with each other to form an interface film, indicating the strong physicochemical stability and maintained integrity of OBs under environmental stress (such as drying and freezing) [49,50]. The particle size of the oil droplets of the peanut and reconstituted OBs prepared with the protein–phospholipid complex was similar: proteins and phospholipids were evenly distributed on the surface of oil droplets, and there were no obvious differences in microstructure (Figure 9A–C,H–J).

### 3.8. Exploration of Stability Mechanism of Peanut OBs

In order to study the stability mechanism of peanut OBs, we explored the internal relationships among OB protein and phospholipid interactions in the aqueous phase, the oil–water interface behavior, and the stability of reconstituted OBs from the perspectives of bulk phase, interface, and macro angle. The OB protein–phospholipid complex system showed an obvious synergic effect in increasing interface pressure (Figure 5), which was mainly attributed to the combination of OB proteins and phospholipids through hydrophobic and electrostatic interactions in aqueous solution before adsorption to the oil–water interface (Figure 2, Figure 3 and Figure 4). Interactions between OB proteins and phospholipids led to conformational changes in proteins, which was conducive to the further extension of protein molecules at the oil–water interface, resulting in an increase in interfacial pressure and increased rate of interfacial pressure of the OB protein–phospholipid complex system, being significantly higher than those of OB proteins alone. The results of *E*-t, *E*-*π*, *E_d_*-t and tan θ (Figure 7) showed a good correlation, revealing that OB proteins and phospholipids adsorbed at the oil–water interface, with strong interactions between the components of adsorption layer, and interaction gradually increased with the increase in adsorption time. The interfacial film formed by OB proteins and phospholipids at the oil–water interface was mainly elastic, and the elastic behavior gradually increased with the increase in adsorption time, showing steady-state elastic behavior. Proteins can form highly elastic interfacial films, and charged proteins adsorbed on the surface of oil droplets generate electrostatic repulsion between oil droplets, providing electrostatic stability for emulsions [51,52,53]. The oil droplets of reconstituted OBs prepared with OB proteins alone were spherical, and OB proteins were evenly distributed on the surface of the oil droplets (Figure 9F,G). Previous studies showed that an appropriate amount of small-molecule surfactant could improve the adsorption state of protein at the oil–water interface, enhance the ability of the protein interface membrane to resist deformation, increase the stability of the protein stabilization system to a certain extent, and avoid oil droplet aggregation [54,55]. Phospholipids promoted the adsorption of OB proteins at the oil–water interface, resulting in an increase in interfacial pressure, which reduced the particle size of reconstituted OBs prepared by OB proteins and increased its stability (Figure 8). Phospholipids increased the net charge of reconstituted OBs prepared by OB proteins, increased electrostatic repulsion and the stability of reconstituted OBs, and gradually increased the stability with the increase in phospholipid content.

According to the internal relationships among the OB protein and phospholipid interactions in the aqueous phase, the oil–water interface behavior, and the stability of reconstituted OBs, a stability mechanism of peanut OBs was proposed. OB proteins and phospholipids were bound through hydrophobic and electrostatic interactions, resulting in conformational changes in OB proteins, which promoted the adsorption of OB proteins and phospholipids at the oil–water interface. OB proteins and phospholipids showed an obvious synergistic effect in increasing interface pressure, and the elastic behavior of the interface was relatively stable, which was beneficial to the formation and stability of the oil–water interface. OB proteins and phospholipids are important components of OBs, and also important factors affecting the stability of OBs. Phospholipids were bound to OB proteins through hydrophobic and electrostatic interactions. The addition of phospholipids increased the net charge on the reconstituted OBs’ surface, enhanced electrostatic repulsion, reduced particle size, and improved the physicochemical stability of reconstituted OBs.

## 4. Conclusions

According to the internal relationships among the OB protein and phospholipid interactions in the aqueous phase, the oil–water interface behavior, and the stability of reconstituted OBs, a novel stability mechanism of peanut OBs is proposed. OB proteins and phospholipids bind through hydrophobic and electrostatic interactions, resulting in conformational changes in proteins, which promote the adsorption of OB proteins and phospholipids at the oil–water interface. OB proteins and phospholipids showed an obvious synergistic effect in increasing the interface pressure, and the elastic behavior of the interface was relatively stable, which was beneficial to the formation and stability of the oil–water interface. Using OB proteins and phospholipids to reconstitute OBs, phospholipids were bound to OB proteins through hydrophobic and electrostatic interaction, and proteins and phospholipids were uniformly covered on the surface of oil droplets, forming a stable interface film on the OBs’ surface to maintain the stability of OBs. Collectively, our results demonstrate that OB proteins and phospholipids largely contribute to the overall stability of OBs. The addition of phospholipids increased the net charge on the surface of reconstituted OBs, enhanced electrostatic repulsion, reduced the particle size, and improved the overall physicochemical stability. The stable structure of OBs restricts the release of oil, which is a “bottleneck” in the industrial application of the aqueous enzymatic extraction method. Our results suggest that OBs can be demulsified by destroying interactions between proteins and phospholipids. This study provides new insights into demulsification and a theoretical basis for realizing the industrialization of extracting peanut oil using the aqueous enzymatic method.

## Figures and Tables

**Figure 1 foods-12-03446-f001:**
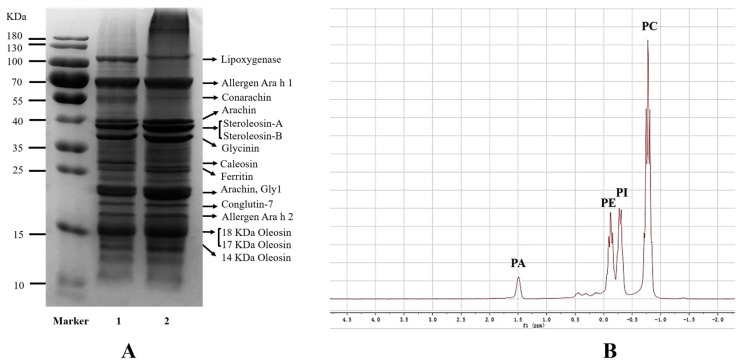
Composition of OB proteins (**A**) (lane 1, peanut OBs; lane 2, OB protein powder) and phospholipids (**B**).

**Figure 2 foods-12-03446-f002:**
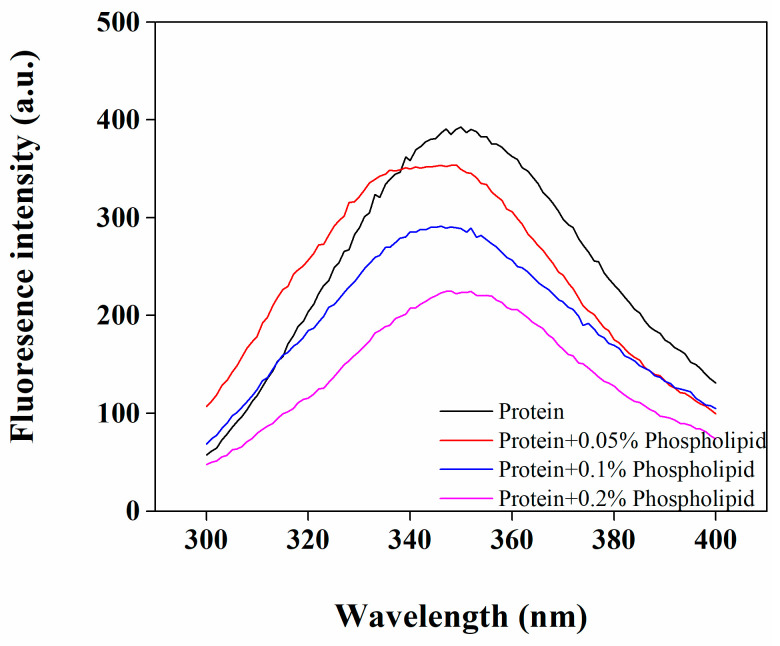
Effect of phospholipids on the endogenous fluorescence spectrum of OB proteins.

**Figure 3 foods-12-03446-f003:**
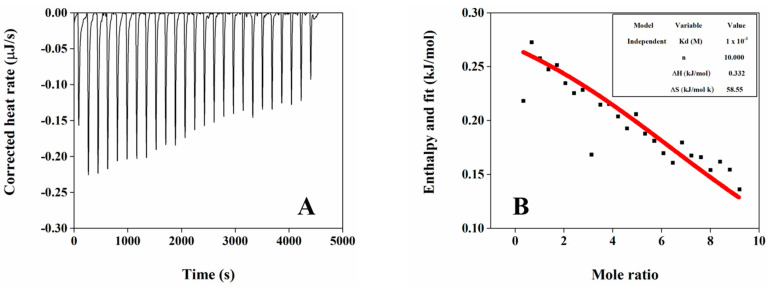
ITC results of interaction between OB proteins and phospholipids: (**A**) titration curve of OB proteins and phospholipids; (**B**) thermodynamic parameters obtained by fitting titration curve.

**Figure 4 foods-12-03446-f004:**
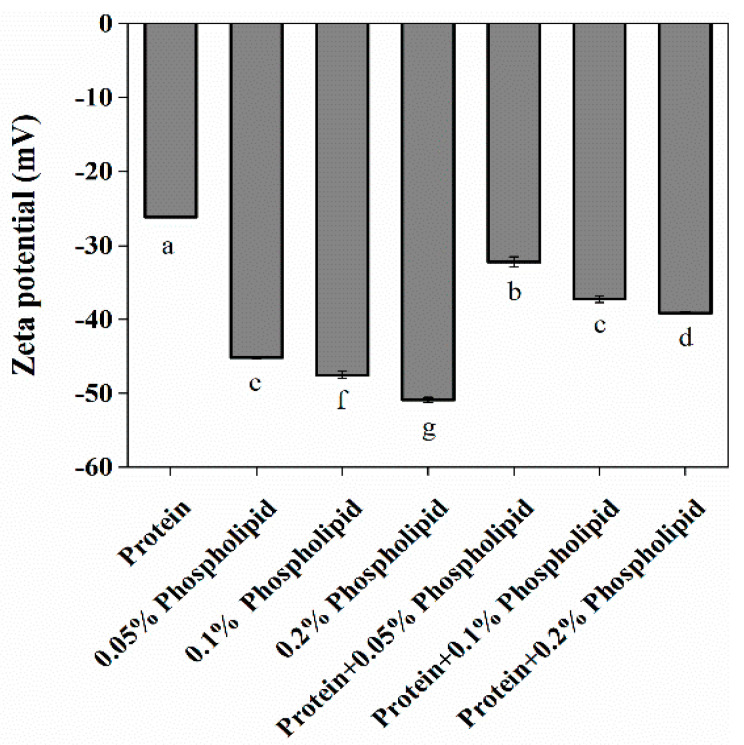
Effect of phospholipids on the zeta potential of OB proteins. Note: Different letters indicate significant differences between groups (*p* < 0.05).

**Figure 5 foods-12-03446-f005:**
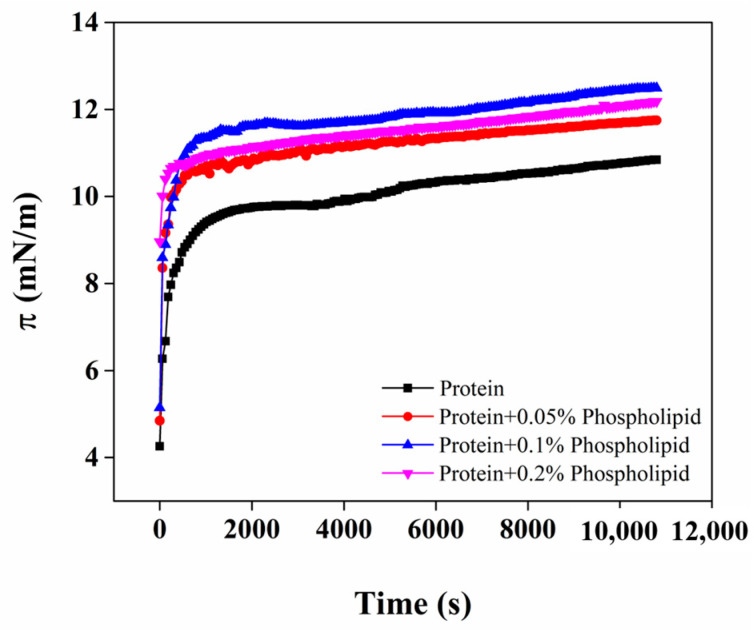
Changes in interfacial pressure of OB proteins and phospholipids at the oil–water interface with adsorption time.

**Figure 6 foods-12-03446-f006:**
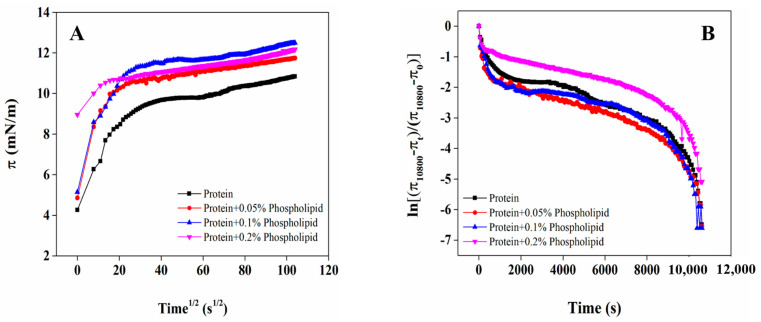
Effect of phospholipids on adsorption behavior of OB proteins at oil–water interface.

**Figure 7 foods-12-03446-f007:**
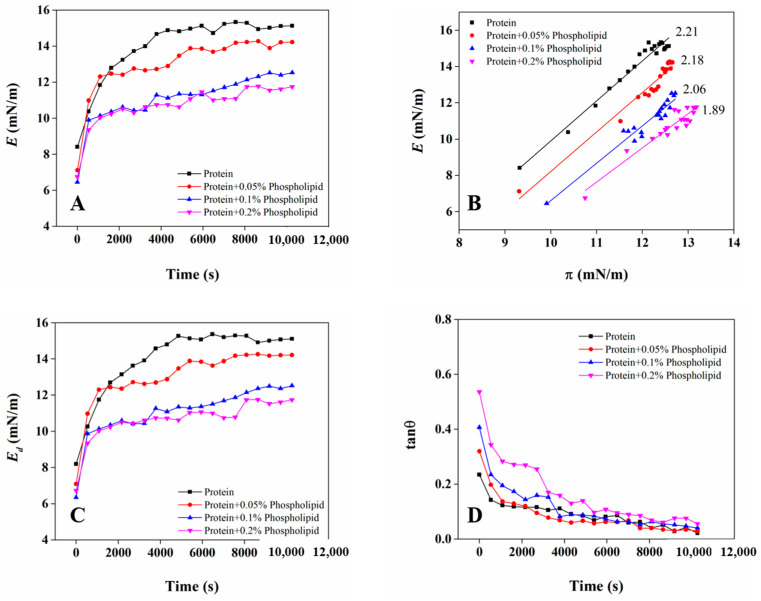
*E* of OB proteins and phospholipids at oil–water interface varies with adsorption time (**A**); *E* varies with *π* (**B**); *E_d_* varies with adsorption time (**C**); tan θ varies with adsorption time (**D**).

**Figure 8 foods-12-03446-f008:**
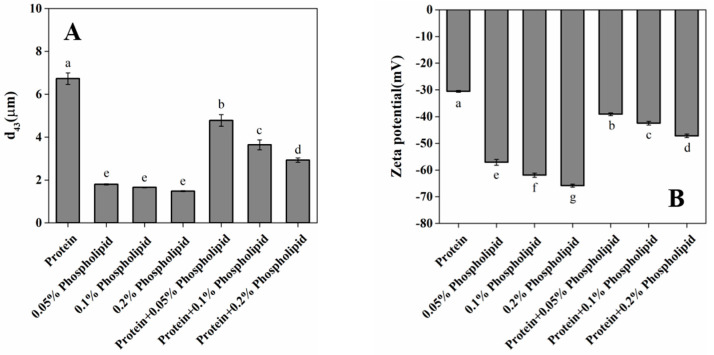
Particle size and zeta potential of reconstituted OBs: (**A**) d_43_; (**B**) zeta potential. Note: Different letters indicate significant differences between groups (*p* < 0.05).

**Figure 9 foods-12-03446-f009:**
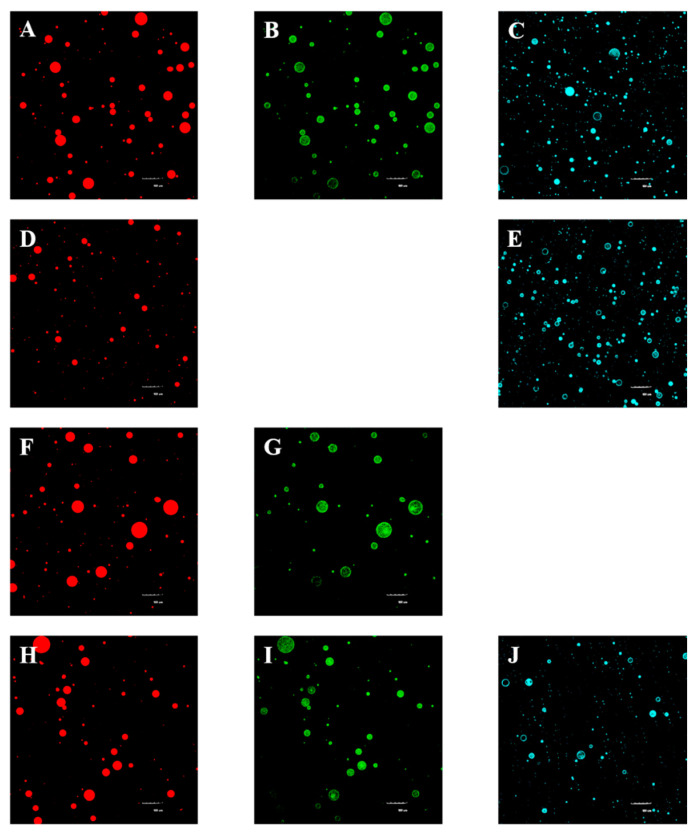
Microstructure of reconstituted OBs: (**A**–**C**) peanut OBs; (**D**,**E**) phospholipid-stabilized OBs; (**F**,**G**) OB-protein-stabilized OBs; (**H**–**J**) reconstituted OBs prepared with OB proteins and phospholipids. (Note: Red is Nile Red-dyed fat, green is FITC-dyed protein, and blue is Rd-DHPE-dyed phospholipid.)

**Table 1 foods-12-03446-t001:** Effects of phospholipids on adsorption kinetics parameters of OB proteins at oil–water interface.

	K_diff_ (mN/m/s^1/2^) (LR)	K_p_ × 10^4^ (s^−1^) (LR)	K_r_ × 10^4^ (s^−1^) (LR)	π_10800_ (mN/m)
Protein	0.2343 ± 0.01 c	2.644 ± 0.05 a	9.911 ± 0.05 a	10.84 ± 0.05 c
Protein + 0.05%Phospholipid	0.3005 ± 0.04 a	1.953 ± 0.04 b	9.203 ± 0.03 b	11.75 ± 0.01 b
Protein + 0.1% Phospholipid	0.2722 ± 0.04 b	1.546 ± 0.03 c	8.958 ± 0.07 c	12.49 ± 0.01 a
Protein + 0.2% Phospholipid	0.1023 ± 0.01 d	1.582 ± 0.02 c	6.284 ± 0.07 d	12.18 ± 0.03 a

Note: Different letters in the same column indicate significant differences between groups (*p* < 0.05).

## Data Availability

The data used to support the findings of this study can be made available by the corresponding author upon request.

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
