# Peer review of "Study on the Stability Mechanism of Peanut OBs Extracted with the Aqueous Enzymatic Method"

_foods, 2023, doi:10.3390/foods12183446_

Round 1
Reviewer 1 Report
The paper explores with adequate techniques the properties of old bodies stabilized with phospholipids. They employ a quite number of well established methodologies to determine surface properties of suspended particles. However, the presentations of the results is confusing and misleading. For example, the PBs proteins of apparently 650 um (according to figure 6) are titrated with phospholipids (figure 2) fluorescence of Tryp is shifted to lower frequencies, implying according to the authors a hydrophobic interaction. however further addition of phospholipids displaces the frequency to higher frequencies restoring that of pure proteins with a lower intensity. This incongruence is not explained,
A similar problem is with figure 5 in which the correlation of surface pressure at long times do not follow a pattern with lipid concentration.
The presentation oof the data of zeta potential is also confusing (fig 4).
I deduce that OBs in buffer has a potential of -25 mV and goes to -31 with 0.05% phospholipid. and so on with the other lipid concentration.
A plot of binding of PL to Ob is desirable.
The bars of pure phospholipids in Figure 4 difficult to understand. Are this zeta potential of pure phospholipids at different concentrations? are them lipids agrégate as vesicles or liposomes? why the zeta potential changes with lipids concentration? different type of particles are formed?
The image in microscopy may help to have an insight of these picture but should be carefully analyzed.
The manuscript should be revised in some verbs times.
Author Response
Dear Editor:
This is regarding manuscript foods-2583752 entitled "Study on stability mechanism of peanut OBs extracted by aqueous enzymatic method" submitted to the Foods. Thanks for your comments, and the modified content in revised manuscript have been highlighted in red.
Replies to the comments of the reviewer(s) are as follows:
Reviewer 1
Comments and Suggestions for Authors:
- The paper explores with adequate techniques the properties of old bodies stabilized with phospholipids. They employ a quite number of well established methodologies to determine surface properties of suspended particles. However, the presentations of the results is confusing and misleading. For example, the PBs proteins of apparently 650 um (according to figure 6) are titrated with phospholipids (figure 2) fluorescence of Tryp is shifted to lower frequencies, implying according to the authors a hydrophobic interaction. however further addition of phospholipids displaces the frequency to higher frequencies restoring that of pure proteins with a lower intensity. This incongruence is not explained.
Thanks for your comment. When additive content of phospholipid increased from 0% to 0.2%, maximum fluorescence peak showed slight blueshift from 350 nm to 348 nm, indicating hydrophobicity of microenvironment around tryptophan residue on protein peptide chain increased, possibly due to interaction between tryptophan and hydrophobic part of phospholipid. Maximum fluorescence peak was 349, 348, and 348 nm when additive content of phospholipid was 0.05%, 0.1%, and 0.2%, respectively. The intuitive effect of Figure 2 might mislead you. There was no inconsistency as you mentioned above.
- A similar problem is with figure 5 in which the correlation of surface pressure at long times do not follow a pattern with lipid concentration.
Thanks for your comment. It could be seen from Figure 5 that the correlation of surface pressure at long times do not follow a pattern with phospholipid concentration indeed. Adsorption of proteins and phospholipid from bulk phase to oil-water interface involves three main steps: diffusion, permeate, and rearrangement. Adsorption process was complex, which might be affected by interaction between proteins and phospholipid, viscosity of system, and other reasons. Although it was inconsistency between surface pressure result and phospholipid concentration, result of Figure 5 indicated OBs protein and phospholipid were gradually adsorbed on oil-water interface and had a synergistic effect in increasing interfacial pressure.
- The presentation of the data of zeta potential is also confusing (fig 4).
Thanks for your comment. Fig.4 reflected effects of phospholipid on zeta potential of OBs protein solution. OBs protein solution and phospholipid solution were negatively charged under neutral conditions, and absolute value of zeta potential of phospholipid solution increased significantly with the increase of phospholipid concentration. OBs protein were negatively charged under neutral conditions, but amphiphilic fragments of oleosin and caleosin contained relatively rich basic amino acids, and they could interact with negatively charged groups of phospholipids. Phospholipid interact with OBs protein, which changed protein conformation, resulting in zeta potential change of OBs protein solution.
- I deduce that OBs in buffer has a potential of -25 mV and goes to -31 with 0.05% phospholipid. and so on with the other lipid concentration.
Thanks for your comment. In this study, zeta potential of reconstituted OBs stabilized by OBs protein was -30.54 mV. Zeta potential of reconstituted OBs prepared by OBs protein - 0.05%phospholipid, OBs protein - 0.1%phospholipid, and OBs protein - 0.2%phospholipid was -39.14, -42.48, and -47.20 mV, respectively. Difference between the test results and results you deduced might be due to adsorption of protein and phospholipid at oil-water interface, as well as interactions between protein and phospholipid.
- A plot of binding of PL to Ob is desirable.
Thanks for your comment. Protein and phospholipid are important components of OBs, which are also important factors to maintain its stability. In this study, stability mechanism of OBs was discussed by systematically analyzed internal relationships among OBs protein and phospholipid interaction in aqueous phase, oil-water interface behavior, and stability of reconstituted. Since phospholipids were just one of study focuses, a plot of binding of PL to Ob were not required.
- The bars of pure phospholipids in Figure 4 difficult to understand. Are this zeta potential of pure phospholipids at different concentrations? are them lipids agrégate as vesicles or liposomes? why the zeta potential changes with lipids concentration? different type of particles are formed?
Thanks for your comment. The bars of pure phospholipids in Fig.4 reflected zeta potential of pure phospholipids solution at different concentrations. OBs protein solution, phospholipid solution and OBs protein-phospholipid complex solution were prepared according to section 2.3.1. Lipids agrégate as vesicles or liposomes might be formed during phospholipid solution preparation, which have not been verified in this study, but zeta potential of phospholipid solution system changed with phospholipid concentration. Phospholipid bound to OBs protein, which made protein structure unfold, resulting in zeta potential change of OBs protein solution. Fig.4 reflected effects of phospholipid on zeta potential of OBs protein solution, and purpose of Fig.4 was to investigate interaction between OBs protein and phospholipid in aqueous solution.
- The image in microscopy may help to have an insight of these picture but should be carefully analyzed.
Thanks for your comment, and the analysis of image in microscopy have been improved in revised manuscripts.
- Comments on the Quality of English Language: The manuscript should be revised in some verbs times.
Thanks for your comments, language of the manuscript has been polished professionally, including revise of some verb times.

Reviewer 2 Report
This manuscript provides valuable information on the stability mechanisms studied of peanut OBs, which are a theoretical basis for demulsification.
I recommend complementing your results with spectra not only of phosphorus, but of carbon and hydrogen that allows you to make a better description and analysis of results, which will greatly complete your research question and complement the information already obtained. And the conclusions can be improved with the proposed assays.
The references are appropriate
I consider that some points of the manuscript should be worded better. For example, the way in which the scientific name is described is not the most appropriate. The presentation and description of your figures can be improved.
Author Response
Dear Editor:
This is regarding manuscript foods-2583752 entitled "Study on stability mechanism of peanut OBs extracted by aqueous enzymatic method" submitted to the Foods. Thanks for your comments, and the modified content in revised manuscript have been highlighted in red.
Replies to the comments of the reviewer(s) are as follows:
Reviewer 2
Comments and Suggestions for Authors:
This manuscript provides valuable information on the stability mechanisms studied of peanut OBs, which are a theoretical basis for demulsification.
- I recommend complementing your results with spectra not only of phosphorus, but of carbon and hydrogen that allows you to make a better description and analysis of results, which will greatly complete your research question and complement the information already obtained. And the conclusions can be improved with the proposed assays.
Thanks for your comment. Complementing phosphorus, carbon, and hydrogen spectra experiment can make a better description and analysis of results indeedly, which will greatly complete research question and complement the information already obtained. Since receiving the revision request, I have carried out relevant supplementary experiments, but they have not been completed due to purchase of reagent, failure of sample pretreatment, instruments, and so on. Furthermore, it was a pity that editor require me to submit my revision by 3 September 2023. Therefore, no relevant experimental results were added in revised manuscripts.
- The references are appropriate.
Thanks for your comment.
- I consider that some points of the manuscript should be worded better. For example, the way in which the scientific name is described is not the most appropriate. The presentation and description of your figures can be improved.
Thanks for your comment, and language of the manuscript has been polished professionally. In revised manuscript, some points have been worded better, and the presentation and description of figures have been improved.

Reviewer 3 Report
In this work, the authors evaluate the stability mechanism of peanut OBs extracted by aqueous enzymatic method. The work is technical sound and the e authors have utilized appropriate techniques of analysis. Some sentences are rambling, and some issue must be addressed in the work. The changes are listed before:
- Please in the figure 1, indicate better the position of the different isoform of oleosin, otherwise from the figure it seems below 15 kDa.
- Please increase the quality of the rectangle present in the figure 3 (for example, font size, width size) otherwise it is hard to read the data.
- Discussions are weak, it requires significant improvement to justify the implication of this study’s experimental findings.
- Avoid abbreviations in the abstract.
- The conclusions must be implemented to better explain the potentiality of the work and the future perspectives.
Moderate editing of English language required
Author Response
Dear Editor:
This is regarding manuscript foods-2583752 entitled "Study on stability mechanism of peanut OBs extracted by aqueous enzymatic method" submitted to the Foods. Thanks for your comments, and the modified content in revised manuscript have been highlighted in red.
Replies to the comments of the reviewer(s) are as follows:
Reviewer 3
Comments and Suggestions for Authors:
In this work, the authors evaluate the stability mechanism of peanut OBs extracted by aqueous enzymatic method. The work is technical sound and the e authors have utilized appropriate techniques of analysis. Some sentences are rambling, and some issue must be addressed in the work. The changes are listed before:
- Please in the figure 1, indicate better the position of the different isoform of oleosin, otherwise from the figure it seems below 15 kDa.
Thanks for your comments. The position of the different isoform of OBs protein in Fig. 1A has been changed and indicated better in revised manuscript.
- Please increase the quality of the rectangle present in the figure 3 (for example, font size, width size) otherwise it is hard to read the data.
Thanks for your comments. The quality of the rectangle present in the Fig. 3 has been improved in revised manuscript, making it easy to read the data.
- Discussions are weak, it requires significant improvement to justify the implication of this study’s experimental findings.
Thanks for your comments. Discussions has been improved in the revised manuscripts.
- Avoid abbreviations in the abstract.
Thanks for your comments. Abbreviation of oil bodies is OBs. Oil bodies appeared many times in abstract, which affected language clear and concise of manuscript. Full name of OBs has been indicated when oil bodies first appeared. Therefore, OBs could be used in the abstract.
- The conclusions must be implemented to better explain the potentiality of the work and the future perspectives.
Thanks for your comments. Conclusions has been improved to better explain the potentiality of the work and the future perspectives in the revised manuscripts.
- Comments on the Quality of English Language: Moderate editing of English language required.
Thanks for your comments, language of the manuscript has been polished professionally.
